# Natural Variation of *StNADC* Regulates Plant Senescence in Tetraploid Potatoes (*Solanum tuberosum* L.)

**DOI:** 10.3390/ijms26094389

**Published:** 2025-05-05

**Authors:** Jiaojiao Zhang, Jianfei Xu, Chunsong Bian, Shaoguang Duan, Jun Hu, Junhong Qin, Huan Wu, Ming He, Yinqiao Jian, Yanfeng Duan, Jiangang Liu, Wanxing Wang, Guangcun Li, Liping Jin

**Affiliations:** State Key Laboratory of Vegetable Biobreeding, Key Laboratory of Biology and Genetic Improvement of Tuber and Root Crop of Ministry of Agriculture and Rural Affairs, Institute of Vegetables and Flowers, Chinese Academy of Agricultural Sciences, Beijing 100081, China; zhangjiaojiaocaas@163.com (J.Z.);

**Keywords:** senescence, *StNADC*, NAD, CRISPR/Cas9, tetraploid potatoes

## Abstract

Senescence impacts plant growth and yields in tetraploid potatoes (*Solanum tuberosum* L.). Because of their homogenous tetraploid features, it is a major challenge to understand the genetic basis and molecular mechanisms of senescence. Here, we identified a novel central senescence regulator (Nicotinate-nucleotide pyrophosphorylase QPT/*StNADC*) through map-based cloning. Overexpression of *StNADC*^Z3^ accelerated senescence in the late-senescence variety, with NAD content declining by around 40%. CRISPR/Cas9-induced *StNADC* mutant *cr2–11* exhibited extremely early senescence, and the NAD content was reduced by 87% along with reduced chlorophyll content and photosynthesis. Moreover, the downstream products of the NAD synthesis pathway, such as NaMN, NAD, or niacin, can refresh the *cr2–11* mutant to grow normally. Further, the transcriptomics and metabolomics data unveiled that the disrupting of *StNADC* impairs NAD metabolism, accelerating plant senescence through multiple biological levels. Our results show that *StNADC* is indispensable for NAD synthesis, and targeting the *StNADC*-mediated NAD synthesis pathway could be a useful strategy to regulate senescence in potato breeding preprograms.

## 1. Introduction

Potato (*Solanum tuberosum* L.) is the world’s third most important food crop after rice and wheat (FAO: https://www.fao.org/faostat/en/#compare, accessed on 5 June 2024) and is critical for global food security [1]. Plant senescence, the final stage of plant development, is a genetically complex process regulated by both intrinsic and environmental factors [2]. Plant senescence is a highly regulated process involving the degradation of chloroplasts and macromolecules such as nucleic acids, proteins, and lipids [3,4]. In crops, senescence facilitates the remobilization of nutrients to harvestable organs, directly impacting yield. Therefore, understanding the regulation of leaf senescence is crucial for improving plant adaptability and agricultural productivity [5]. Senescence is governed by multiple layers of regulation, including chromatin remodeling, transcriptional and post-transcriptional control, and translational and post-translational modifications [4,6]. Recent advances in multi-omics approaches have provided a more comprehensive understanding of the spatio-temporal dynamics underlying this complex process [7].

In tetraploid potatoes, senescence is closely linked to yield and adaptability [2]. Rather than being a simple degenerative process, senescence is a developmentally controlled recycling mechanism. During this stage, assimilates and nutrients accumulated during growth are transported from senescing leaves to developing seeds and storage organs, directly contributing to crop yield [3,5]. Therefore, the precise initiation and regulation of leaf senescence are essential for optimal plant growth and tuber formation. By leveraging natural variation in senescence timing, potato breeders can develop varieties suited to different latitudes and harvest schedules, meeting specific market needs [8]. However, the genetic complexity of tetraploid potatoes presents significant challenges in understanding the genetic and molecular mechanisms underlying senescence.

Senescence is marked by a decline in antioxidant capacity, leading to the accumulation of reactive oxygen species (ROS) [9,10]. Oxidative stress (ROS accumulation) is a key driver of senescence. NADPH oxidase (RBOH)-generated H_2_O_2_ (an ROS) regulates senescence in salt-stressed plants, where NAD+ metabolism influences redox homeostasis [9,10]. ROS serve as key regulators of senescence by modulating core transcription factors such as WRKY and NAC, as well as altering the structure and function of critical enzymes [11,12]. A major source of ROS is the electron transport chain associated with nicotinamide adenine dinucleotide (NAD^+^) [13,14]. NAD^+^, a central redox molecule, mediates hundreds of cellular reactions and is indispensable for numerous metabolic pathways [15]. In humans and mice, NAD^+^ depletion has been linked to both health and senescence [16,17,18].

In animals, NAD^+^ is synthesized through three pathways: the de novo pathway, the salvage pathway, and the kynurenine pathway [16,19]. In the de novo pathway, L-aspartate oxidase (AO, EC 1.4.3.16) catalyzes the oxidation of L-aspartate to iminoaspartate, which is then converted to quinolinate by quinolinate synthase (QS). The kynurenine pathway converts tryptophan into quinolinic acid (QA) through a series of enzymatic reactions. In both pathways, QA is catalyzed by quinolinate phosphoribosyltransferase (QPTase, encoded by NADC) to produce nicotinate mononucleotide (NaMN), a precursor for NAD^+^ synthesis [20]. The salvage pathway recycles NAD^+^ degradation products back into NAD^+^ [21]. In plants, NAD+ biosynthesis primarily occurs through the aspartate (de novo) pathway, with a secondary salvage pathway for recycling nicotinamide (NAM) [22]. NADC plays a central role in NAD^+^ synthesis, acting as a rate-limiting enzyme in both the de novo and salvage pathways [23,24]. Though the role of NAD^+^ in animal senescence is well established, there is also growing evidence that NAD+ (nicotinamide adenine dinucleotide) plays a crucial role in plant senescence. In arabidopsis, NAD+ depletion accelerates leaf yellowing (chlorophyll degradation) and senescence-associated senescence gene expression (e.g., SAG12) [25], while the function and molecular regulation of QPTase/*NADC* in plant senescence remain poorly understood.

In this study, we fine mapped a senescence-related locus to a 75 kb region on chromosome V using a population of 588 progeny and identified *StNADC* as the candidate gene. Overexpression of *StNADC*^Z3^, a natural variant cloned from the early-senescence parent Z3, in the late-senescence parent Z19 accelerated senescence. Conversely, CRISPR/Cas9-induced knockout of *StNADC* in Z19 resulted in premature senescence and reduced NAD^+^ levels. Exogenous supplementation with NAD^+^, NaMN, or niacin restored normal growth in *cr2–11* mutants. Transcriptomic and metabolomic analyses of *cr2–11* mutants revealed that disruption of *StNADC* impaired NAD^+^ metabolism, accelerating senescence through multiple biological processes, including transcriptional regulation, protein post-translational modification, nucleic acid metabolism, chlorophyll metabolism, and photosynthesis. These findings demonstrate that *StNADC* regulates potato senescence through the NAD^+^ synthesis pathway.

## 2. Results

### 2.1. Phenotypic Analysis of Plant Senescence in the Segregating Population

To investigate the plant senescence trait in potato, we crossed two tetraploid varieties, the early-senescence parent Zhongshu 3 (Z3) and the late-senescence parent Zhongshu 19 (Z19), generating an F1 segregating population of 588 individuals. Plant senescence was categorized into five scales based on the percentage of senescent tissue (Figure 1A). The F1 population and both parents were evaluated in two locations, Baoding and ChaBei, over three consecutive years (2019, 2020, and 2021). These six environments were designated as 2019BD, 2020BD, 2021BD, 2019CB, 2020CB, and 2021CB, respectively. Across all environments, significant differences in senescence were observed between the two parents. The F1 population exhibited a normal distribution for the senescence trait (Figure 1A). These results indicate that plant senescence in potato is a polygenic trait controlled by quantitative genetic mechanisms.

### 2.2. Fine Mapping of the Senescence-Related QTL

In our previous study, a major Quantitative Trait Locus (QTL) associated with early senescence in potato was identified on the short arm of chromosome V, spanning a 447 kb region [26]. To further refine this region and identify the key gene responsible for senescence, we utilized SNPs from a 20k chip [27] and re-sequencing data of the parental lines Z3 and Z19 to develop KASP markers. A total of five markers were successfully designed (Appendix A). The most informative markers for delimiting the senescence interval were c5_4041510 and c5_4116968 (based on the DM 4.03 genome version) (Figure 1B). Phenotypic analysis of individual progenies using these markers revealed two recombination events in early-senescence progenies and three in late-senescence progenies (Figure 1B and Appendix A). This allowed us to narrow the interval to a 75 kb region at the distal end of chromosome V, containing 11 predicted genes (Figure 1B and Appendix A).

### 2.3. Soltu.DM.05G004780 Is the Candidate Gene for the Plant Senescence Trait

Among the 11 candidate genes, *Soltu.DM.05G004780* encodes a quinolinate phosphoribosyltransferase (QPTase) enzyme, a member of the NADC family. We therefore designated *Soltu.DM.05G004780* as *StNADC* in this study. In humans and mice, depletion of *NADC* has been linked to senescence [16,17,18]. Real-time PCR analysis revealed that *StNADC* is expressed in roots, stems, leaves, and flowers (Appendix A), with significantly higher expression levels in Z19 compared to Z3 during the late stage of leaf development (Appendix A). The coding sequence (CDS) of *StNADC c*omprises 1059 base pairs (bp), encoding 353 amino acids (Appendix A). Sequencing identified three key SNPs (C33T, G297A, and A938G) between Z19 and Z3, resulting in two amino acid changes (H11Y and A99T), while A938G is a non-synonymous mutation (Appendix A). Genotyping of 218 F1 progenies and 99 natural varieties using the KASP marker c5_4116968 confirmed that the SNP G297A is associated with early and late senescence (Figure 1C,D). These results collectively suggest that *StNADC* is a candidate gene regulating plant senescence, with the unique allele in the early-senescence parent Z3 designated as *StNAD^C^*^Z3.^

### 2.4. StNADC^Z3^ Promotes Plant Senescence in Tetraploid Potatoes

To further validate the role of *StNADC* in plant senescence, we generated transgenic lines of the late-senescence parent Z19 overexpressing the *StNADC*^Z3^ allele under the control of the cauliflower mosaic virus 35S promoter (CaMV 35S). Transgenic lines OE-*StNADC*1–2 and OE-*StNADC*1–4 exhibited earlier leaf yellowing compared to the wild type (Figure 1E). High expression levels of *StNADC* in these lines were confirmed by qPCR (Figure 1F). Additionally, NAD content in the transgenic lines decreased by approximately 40% compared to the wild type (Figure 1G), and the photosynthetic rate was also reduced (Figure 1H). These findings indicate that overexpression of *StNADC*^Z3^ induces early senescence, accompanied by reduced NAD levels and impaired photosynthesis. Subcellular localization analysis revealed that *StNADC* is localized in the chloroplasts of *Nicotiana benthamiana* leaves (Appendix A).

### 2.5. CRISPR/Cas9-Induced Mutation of StNADC Triggers Early Senescence

To further investigate the role of *StNADC* in plant senescence, we designed a CRISPR/Cas9 construct targeting the first exon of *StNADC*. A single guide RNA (sgRNA) was designed to target a conserved sequence (cs) of *StNADC*, and the construct was introduced into the late-senescence parent Z19. Among the resulting mutants, one was nonviable due to a complete loss of *StNADC* function. However, a viable mutant, *cr2–11* (referred to as cr-*StNADC*), which retains partial *StNADC* function, was successfully obtained (Figure 2A,B). Sequencing analysis revealed that *cr2–11* carries a 56 bp deletion and a 7 bp insertion in the *StNADC* gene (Figure 2A).

Compared to the wild type, the *cr2–11* mutant exhibited pronounced dwarfism and early senescence (Figure 2B–D). Quantitative PCR confirmed a significant reduction in *StNADC* expression in *cr2–11* (Figure 2E). Physiological analyses conducted 30 days after propagation revealed an 87% reduction in NAD content in *cr2–11* relative to the wild type (Figure 2F). Additionally, chlorophyll a, chlorophyll b, and total chlorophyll levels were markedly lower in the mutant (Figure 2G). In contrast, reactive oxygen species (ROS) levels in *cr2–11* were approximately 1.4-fold higher than in the wild type (Figure 2H). These results demonstrate that partial loss of *StNADC* function induces early senescence, accompanied by a significant decline in NAD and chlorophyll levels, as well as elevated ROS accumulation. These findings underscore the critical role of *StNADC* in regulating senescence and maintaining cellular homeostasis in potato.

### 2.6. NAD and Its Intermediates Rescue the cr2–11 Mutant Phenotype

The *StNADC* mutant *cr2–11* exhibits early senescence, leading to premature death. To investigate the role of *StNADC* in the NAD biosynthesis pathway, we examined whether supplementation with NAD or its intermediates could rescue the *cr2–11* phenotype. Wild-type and *cr2–11* plants were cultured on MS medium supplemented with downstream products of *StNADC*, including NAD, nicotinate mononucleotide (NaMN), and niacin [24]. The results demonstrated that the growth of *cr2–11* was fully restored by the addition of NAD at concentrations ranging from 0.01 g/L to 0.1 g/L, while 0.001 g/L NAD partially restored growth (Figure 3A). Similarly, NaMN at a concentration of 0.001 g/L partially rescued the *cr2–11* mutant phenotype (Figure 3B). Niacin supplementation fully restored growth at concentrations of 0.2 g/L to 0.4 g/L, while 0.1 g/L niacin resulted in partial restoration (Figure 3C). The ability of niacin, a substrate of the salvage pathway, to rescue the *cr2–11* mutant underscores the critical role of the salvage pathway in NAD synthesis. These findings highlight the importance of *StNADC* in the NAD biosynthesis pathway and establish it as a key target for manipulating NAD levels in plants.

### 2.7. Analysis of NAD Metabolic Networks in cr2–11 Mutants

To elucidate the regulatory network of *StNADC*, we performed transcriptomic and metabolomic analyses on *cr2–11* and wild-type plants at two developmental stages: the young stage (Day 15, healthy phase) and the older stage (Day 30, senescence phase). Metabolomic profiling identified 318 annotated metabolites, primarily lipids (34.65%), amino acids and derivatives (29.13%), and nucleotides and derivatives (12.60%) (Appendix A). Transcriptomic analysis revealed 1735 upregulated and 1223 downregulated genes at Day 15, and 2088 upregulated and 1605 downregulated genes at Day 30 in *cr2–11* compared to the wild type (Appendix A). The greater transcriptional changes occurred at Day 30. qRT-PCR validation of 15 randomly selected DEGs confirmed the reliability of the RNA-seq data (Appendix A).

Further analysis of the NAD biosynthesis pathway revealed that most genes were downregulated in *cr2–11*, accompanied by significantly reduced levels of downstream products NaMN and NAD (Figure 4B,C). Upstream substrates, such as L-Aspartic acid and L-Asparagine (de novo pathway) and L-Tryptophan (kynurenine pathway), accumulated at higher levels in *cr2–11*, indicating impaired NAD synthesis. Additionally, key genes in the salvage pathway (*NMNAT*, *URH1*, *NIC1*, and *NapRT2*) were downregulated, further supporting the central role of *StNADC* in NAD metabolism. These findings demonstrate that *StNADC* is a pivotal regulator of NAD biosynthesis, integrating the de novo, kynurenine, and salvage pathways to maintain cellular homeostasis and delay senescence.

### 2.8. The StNADC Mutation Alters Metabolite Profiles Associated with Early Senescence

Given that alterations in NAD^+^ levels can significantly impact cellular metabolism [25], we analyzed metabolite profiles in the *cr2–11* mutant and wild-type plants. At Day 30, sugar levels, including Glucose-6-phosphate, D-Fructose, and Raffinose levels, were significantly higher in *cr2–11* compared to the wild type (Figure 5A). Elevated sugar levels are a known precursor to plant senescence [26,27], suggesting that the *cr2–11* mutation triggers early changes in sugar metabolism characteristic of senescence. Sugars are essential for generating pyruvate through glycolysis, which fuels mitochondrial respiration via the tricarboxylic acid (TCA) cycle [28]. Analysis of TCA cycle intermediates revealed that L-Malic acid and α-ketoisovaleric acid were reduced in *cr2–11*, while other intermediates remained unchanged (Figure 5B). Additionally, L-proline and L-methionine, derived from α-ketoglutarate, accumulated at higher levels in *cr2–11* at both Day 15 and Day 30 (Figure 5B). These findings suggest a disruption in TCA cycle flux and amino acid metabolism in the mutant. Further evidence of senescence-associated metabolic changes was observed in the accumulation of DNA breakdown intermediates, such as cyclic AMP, GMP, and 2,3-Cyclic CMP, in *cr2–11* at Day 30 (Figure 5C), indicating significant nucleic acid degradation. Additionally, the plant hormone salicylic acid (SA) was elevated in *cr2–11* at Day 30 (Figure 5C), further supporting the onset of senescence.

### 2.9. StNADC Regulates Plant Senescence Through Multiple Layers

To comprehensively analyze the co-expression patterns of key genes and physiological traits affected by *StNADC* knockout, we performed weighted gene co-expression network analysis (WGCNA). The results revealed that NaMN, chlorophyll, α-ketoisovaleric acid, L-malic acid, nicotinic acid, and NAD exhibited similar correlation patterns, showing positive associations with the ME green module (Figure 5D and Appendix A). These metabolites, particularly chlorophyll and NAD, displayed a declining trend during senescence in the *cr2–11* mutant. KEGG pathway analysis of differentially expressed genes (DEGs) in the ME green module highlighted enrichment in translation and energy metabolism, with significant impacts on photosynthesis-related processes, including porphyrin and chlorophyll metabolism, photosynthesis-associated proteins, and carbon fixation (Figure 5E and Appendix A).

In contrast, a second group of metabolites, including L-methionine, tryptophan, glucose-6-phosphate, D-glucosamine-6-phosphate, raffinose, ROS, salicylic acid, abscisic acid, GMP, cyclic AMP, and 2,3-cyclic CMP, were positively correlated with the ME blue module (Figure 5D and Appendix A). These metabolites, particularly ROS, salicylic acid, and abscisic acid, increased during senescence in the *cr2–11* mutant and are known accelerators of plant senescence. The ME blue module was upregulated in the *cr2-11-2* group (Appendix A), suggesting its role in senescence progression. Functional enrichment analysis of the blue module genes revealed associations with autophagy, plant–pathogen interactions, glutathione metabolism, tyrosine metabolism, plant hormone signaling, DNA repair, the ubiquitin system, and protein recombination (Appendix A). Notably, DEGs related to the oxidative stress response, defense response, and DNA repair were significantly upregulated in *cr2–11* (Figure 5F), indicating that *StNADC* deficiency promotes DNA damage and genomic instability. These findings demonstrate that *StNADC* regulates senescence through multiple layers, including sugar accumulation, DNA instability, reduced chlorophyll content, and impaired photosynthesis, all of which are critical for potato growth and development.

## 3. Discussion

Genetic Basis of Senescence in Potato. The genetic regulation of potato senescence has been a focus of research for decades. Early studies identified the El (*Early Inactivation*) locus as a major Quantitative Trait Locus (QTL) for senescence, located on multiple chromosomes (II, IV, V, VII, X, XI, and XII) and associated with traits such as plant height, tuber formation, and disease resistance [28,29]. However, the primary QTL for maturity is consistently mapped to chromosome V, suggesting polygenic control of senescence [30,31]. Despite these advances, only a few senescence-related genes, such as *StCDF1* [8] and *StABI5* [32], have been cloned in potatoes. The molecular mechanisms underlying senescence in tetraploid potatoes remain largely unexplored. In this study, we fine mapped a senescence locus to a 75 kb region on chromosome V, distinct from previously reported loci such as *StCDF1*. This interval contains *StNADC*, a key gene regulating senescence through the NAD biosynthesis pathway.

*StNADC*: A Central Regulator of Senescence via NAD Metabolism. NAD is a critical cofactor in cellular redox reactions, and its homeostasis is essential for preventing cell death and senescence [33]. Here, we identified *StNADC* as a central regulator of senescence in tetraploid potatoes. *StNADC* encodes nicotinate-nucleotide pyrophosphorylase (QPTase), a key enzyme in the de novo NAD biosynthesis pathway. Our study identified three key SNPs (C33T, G297A, and A938G) in *StNADC* between Z3 and Z19, resulting in two amino acid substitutions (H11Y and A99T) and a non-synonymous mutation (A938G). The H11Y substitution (C33T) replaces a positively charged histidine with polar tyrosine in the N-terminal domain, likely disrupting α-helix hydrogen bond networks critical for structural stability and potentially altering substrate/cofactor electrostatic interactions, as observed in *Bacillus subtilis* NADH dehydrogenase variants [34]. The A99T substitution (G297A) introduces a hydroxyl group near the active site, potentially rigidifying the active site loop through new hydrogen bonds while possibly causing steric hindrance to substrate binding, similar to Pseudomonas NAD kinase mutations [35]. The *StNADC*^Z3^ allele accelerates senescence, while knockout or overexpression of *StNADC* leads to NAD deficiency and shortened lifespan. These findings highlight the conserved role of NAD in plant senescence and demonstrate that *StNADC* is indispensable for normal growth. Similar roles for NAD metabolism have been reported in other species. For instance, overexpression of the *Escherichia coli NADC* gene in Arabidopsis increases NAD levels and induces senescence-related genes such as *SAG20*, *SAG13*, and *SAG24* [25]. In rice, mutations in *OsNaPRT1*, a key enzyme in the NAD salvage pathway, result in dwarfism, wilting, and premature senescence.

Multi-Layered Regulation of Senescence by *StNADC*. *StNADC* impacts senescence through multiple interconnected pathways (graphical abstract). NAD deficiency caused by *StNADC* dysfunction disrupts chlorophyll synthesis, photosynthetic efficiency, and energy metabolism, including glycolysis, the pentose phosphate pathway, and the TCA cycle. Additionally, the accumulation of upstream substrates such as L-Asp and Try suggests impaired NAD biosynthesis, which may contribute to DNA damage—a hallmark of senescence [36,37]. Elevated ROS levels in *cr2–11* mutants further accelerate senescence by degrading macromolecules such as lipids and proteins, activating autophagy and ubiquitin-mediated protein degradation pathways [9,38]. Plant hormones also play a critical role in senescence regulation. Abscisic acid (ABA) and salicylic acid (SA) are key regulators of stress responses and developmental processes. In *cr2–11* mutants, the accumulation of ABA and SA, along with the upregulation of *StABI5* and *NBR1*, highlights their involvement in senescence. These hormonal changes, combined with metabolic shifts, lead to nutrient remobilization and the accumulation of soluble sugars such as glucose-6-phosphate and raffinose. Elevated sugar levels are a hallmark of early senescence and support mitochondrial respiration through glycolysis and the TCA cycle [39,40].

Broader Implications and Future Directions. Senescence is a polygenic trait regulated by complex, interconnected pathways. While our study focuses on key candidate gene *StNADC*, numerous additional genetic contributors remain to be functionally characterized—a critical direction for future research. Our findings have broad implications for understanding the regulation of senescence in plants. The identification of *StNADC* as a key regulator of NAD metabolism and senescence provides a new perspective on the role of NAD in plant development and stress responses. The multi-layered regulation of senescence by *StNADC* underscores the complexity of this process and highlights the need for integrated approaches to study senescence in crops. Future research should focus on several key areas: (1) elucidating the molecular mechanisms by which *StNADC* regulates NAD metabolism and senescence; (2) exploring the interactions between *StNADC* and other senescence-related genes and pathways; (3) investigating the role of NAD metabolism in stress responses and adaptation to changing environmental conditions; and (4) developing strategies to manipulate *StNADC* and NAD metabolism to improve crop yield and stress tolerance. These efforts will not only advance our understanding of plant senescence but also contribute to the development of more resilient and productive crop varieties.

## 4. Materials and Methods

### 4.1. Plant Materials and Growth Conditions

For fine mapping, two tetraploid potato varieties, Zhongshu 19 (Z19) and Zhongshu 3 (Z3), along with a segregating population of 588 F1 progenies, were cultivated in two locations in China: ChaBei (CB, 41°41′ N, 114°94′ E, 1450 m above sea level) and Baoding (BD, 38°87′ N, 115°47′ E, 22–23 m above sea level). Due to the heterozygous tetraploid nature of potato parental lines (2n = 4x = 48), the F1 progeny of potato crosses exhibit segregation patterns comparable to the F2 generation in inbred diploid crops (e.g., rice, maize). The plants were grown during the growing seasons of 2019, 2020, and 2021 under standard field conditions. The genetic background of the plant materials, including their agronomic characteristics, was described in detail in our previous study [41].

### 4.2. Design of Field Experiments and Senescence Assessment

Field experiments were designed as randomized augmented trials. Each progeny line was represented by four plants, arranged with a sowing length of 1 m, row spacing of 0.9 m, and plant spacing of 0.25 m. The experiments were conducted in two distinct ecological regions: ChaBei (CB) and Baoding (BD). In ChaBei, sowing occurred in early May, and harvesting took place at the end of September each year. This region exhibits environmental characteristics including gradually decreasing temperatures and shorter daylight hours during the senescence period, which are conducive to tuber formation. In Baoding, sowing was performed in early March, and harvesting occurred in mid-June. During the harvest period, Baoding experiences gradually increasing temperatures and longer daylight hours, which are less favorable for tuber formation. Senescence progression was assessed for each genotype from emergence until 50% of the foliage exhibited senescence. The following scale was used to evaluate senescence: 5: dark-green plants (no senescence); 4: light-green plants (early signs of senescence); 3: 25% of foliage senesced; 2: more than 50% of foliage tissue senesced; 1: dead plant.

### 4.3. Development of KASP Markers for Fine Mapping of Plant Senescence

In addition to SSR and CAPS markers, we developed Kompetitive Allele-Specific PCR (KASP) markers for genotyping and fine mapping. The two parental lines, Z3 (early senescence) and Z19 (late senescence), were genotyped using a potato 20k SNP array, as described in our previous study [27]. Allele discrimination outputs from each KASP assay were used to assign graphical genotypes, where the ‘A’ allele (blue) corresponded to the early-senescence parent (Z3), and the ‘a’ allele (gray) corresponded to the late-senescence parent (Z19). A total of five KASP markers were designed and used to genotype a population of 218 F1 progenies and 99 natural varieties with extreme senescence phenotypes. Phenotypic screening for senescence was conducted, and progeny lifespan was calculated as the number of days from emergence to the onset of senescence. The statistical significance of each marker’s association with senescence was evaluated using a T-test. Detailed information on the KASP markers, including primer sequences and genomic positions, is provided in Appendix A.

### 4.4. Plasmid Constructs

The coding region of *StNADC* was amplified from complementary DNA (cDNA) of the early-senescence parent Z3 and cloned into the *pBI121* plasmid to generate the overexpression vector. For CRISPR/Cas9-mediated gene editing, a 19-nucleotide single guide RNA (sgRNA) targeting the conserved sequence of *StNADC* in the late-senescence parent Z19 was designed and incorporated into the CRISPR-Cas9 vector *pKSE402* [42]. Detailed information on the primers used for gene cloning and CRISPR/Cas9 vector construction is provided in Appendix A.

### 4.5. Generation of Transgenic Lines

The tetraploid potato variety Z19 was used for overexpression and gene editing. Three-week-old plants were used for transformation. Transgenic lines were generated using *Agrobacterium tumefaciens* as described in an earlier study [43]. The positive transformants were screened based on their growth on the medium containing 50 mg/L kanamycin. To detect mutations in *StNADC*, full-length *StNADC* sequences were amplified and sequenced from all regenerated plantlets. Furthermore, PCR products were linked to the T vector (Vazyme pEM-T Vector), and at least 24 monoclonal (tetraploid potatoes) were chosen for variant site analysis.

### 4.6. Maintenance of Transformed and Non-Transformed Controls

Plants were propagated in vitro using single-node stem segments cultured on Murashige and Skoog (MS) medium supplemented with 3% sucrose. Cultures were maintained under long-day conditions (16 h light/8 h dark) at 22 °C, with a light intensity of 2500 µmol m^−2^ s^−1^. Four-week-old wild-type and transgenic plantlets were transplanted into plastic pots (10 cm diameter) at a density of one plant per pot, with nine replicates per genotype. Plants were grown in a greenhouse under long-day photoperiod conditions (16/8 h light/dark) at 20 ± 2 °C to ensure optimal growth conditions. Senescence was quantified as the number of days from planting until 50% of the plants exhibited wilting and yellowing. Plant height was measured from the soil surface to the apical tip 60 days after planting. Tuber formation and fresh weight per plant were evaluated at harvest (90–100 days after transplantation) using at least five plants per genotype.

### 4.7. Restoration of the cr2–11 Mutant Phenotype

To assess the restoration of the *cr2–11* mutant phenotype, nodes from 15-day-old wild-type and *cr2–11* mutant plants grown on MS medium were transferred to fresh MS medium supplemented with varying concentrations of NAD, nicotinate mononucleotide (NaMN), and niacin (Na). The concentrations used were as follows:

NAD: 0.001 g/L, 0.01 g/L, and 0.1 g/L;

NaMN: 0.00005 g/L, 0.0001 g/L, and 0.001 g/L;

Niacin (Na): 0.1 g/L, 0.2 g/L, and 0.4 g/L.

Plant growth was monitored and photographed after two weeks. Each treatment was performed with three independent biological replicates, each consisting of at least 18 plants.

### 4.8. RNA Isolation and Quantitative PCR

The fifth fully expanded leaf from the shoot apex was harvested at 15, 30, 45, 60, and 75 days after planting for gene expression experiments on different leaf ages (DAP). RNA was isolated from the leaves of plants grown under long-day conditions for gene expression analysis of the transformed plants and non-transformed controls. The samples were immediately frozen in liquid nitrogen and kept at –80 °C. The total RNA was isolated according to the manufacturer’s protocol using a polyphenol plant total RNA Kit (Gene Better R318–50). A HiScript III 1st Strand cDNA Synthesis Kit (+gDNA wiper) was used to reverse transcribe the RNA (Vazyme-R312–01). The primer blast website (https://www.ncbi.nlm.nih.gov/tools/primer-blast (accessed on 5 June 2024)) was used to design the gene-specific primers. Real-time PCR was performed with a Taq Pro Universal SYBR qPCR Master Mix (Vazyme-Q712–02/03) and 10 µL of reaction solution containing 1 µL of 10-fold diluted cDNA, according to the manufacturer’s instructions. A Roche 480 was used to measure real-time gene expression. The internal control was the EIF3e gene [8]. The detailed primer information for qPCR is shown in Appendix A. Plant senescence and tuberization-related gene primers are shown in Appendix A. The relative gene expression levels were calculated using the 2−ΔCt method [44].

### 4.9. Determination of Chlorophyll and ROS Contents

For the analysis of chlorophyll content, the samples were incubated in N, N-dimethylformamide at room temperature for 3–4 h in the dark. The chlorophyll content was measured spectrophotometrically at 647 and 664 nm. The samples were incubated with 0.9 NaCl for ROS extraction using an ROS enzyme-linked immunoassay kit (JLC11460) according to the manufacturer’s instructions.

### 4.10. Measurement of NAD

NAD was extracted and assayed according to the protocol of the coenzyme I NAD (H) content detection kit (Solarbio BC0315). A microplate reader (BIO-RAD iMarkTM Microplate Reader, Hercules, CA, USA) was used to determine the A value of each sample measurement tube and control tube, and the following formula was obtained: △A value = A determination −A control. The NAD+ content is further calculated as follows: NAD+(nmol/g) = 1.25 × △A determination ÷ △A standard ÷ W(g).

### 4.11. RNA Sequencing

For the RNA-Seq analysis, 3-week-old (WT-Z19 and *cr2–11*) plants were sampled and homogenized for RNA extraction. Each sample received three biological replicates. On a HiSeq X 10 sequencer, the 200–500 bp library products were enriched, quantified, and finally sequenced (Illumina, San Diego, CA, USA). Final libraries were prepared in a volume of 20 μL and diluted to a standardized concentration of 2 nM using EB buffer (10 mM Tris-HCl, pH 8.0) for sequencing. The raw data were cleaned (filtering out the adaptor and low-quality reads) by Trimmomatic (version 0.36), and UMI reduplication of clean reads (in-house software) was carried out, after which the reduplicated consensus sequences were cleaned again by fast 0.20 to filter out reads shorter than 20 nt and used for RNA-Seq analysis. The clean reads were mapped to the potato *S. tuberosum* Group Phureja DM 1–3 516 R44 (v6.1) reference genome obtained from http://solanaceae.plantbiology.msu.edu/dm_v6_1_download.shtml (accessed on 5 June 2024) [45] and DM v8.1 reference genome, which was published recently at http://www.bioinformaticslab.cn/pubs/dm8/ (accessed on 5 June 2024), respectively [46]. The DEGs were identified (FDR ≤ 0.05 and absolute of log2 fold change ≥ 0.75), the results were similar, and the Gene IDs of the two genomes related to this research are both listed in Appendix A and Appendix A. TBtools was implemented for the Gene Ontology (GO) enrichment analysis (*p*-value ≤ 0.05) of differentially expressed genes (DEGs) and heat map drawing (version v1.0.98) [47].

### 4.12. Determination of Metabolites

The leaf samples for the metabolite analysis were collected at the same time as those for RNA-seq. They were immediately frozen in liquid nitrogen and stored at –80 °C until further analysis. The extraction was performed by rapidly grinding the sample in methanol at room temperature and then ultrasonicating it for 15 min. After that, 200 µL supernatant was taken after centrifugation and detected using a mass spectrometer (Thermo QE-HF-X and Liquid chromatography Thermo Vanquish). The relative levels of metabolites were determined using an established liquid chromatography technique coupled with tandem mass spectrometry, as previously described [48]. The data were normalized using XCMS, as described in earlier studies [49]. Sample extraction, metabolite identification, and quantification were performed by PANOMIX (Suzhou, China).

### 4.13. Co-Expression Network Analysis

WGCNA was used to reconstruct gene modules with different expression patterns from prefiltered expression data from 12 samples. The number nine was chosen as the appropriate soft-thresholding power for a signed network based on the criterion of approximate scale-free topology. With a minimum module size of 10 and a dissimilarity of 0.15, similar expression profiles were merged into the same module utilizing TBtools [47].

### 4.14. Statistics

For comparison of individual treatments with their relevant controls, Unpaired T-tests were used. To compare measurements of multiple treatments with one another, we performed Ordinary One-Way ANOVA followed by multiple pairwise comparisons to determine group differences using GraphPad Prism version 9. **** *p* < 0.0001, *** *p* < 0.001, ** *p* < 0.01, and * *p* < 0.05 for both the *t*-test and one-way ANOVA.

## 5. Conclusions

Our study demonstrates that *StNADC* regulates senescence in tetraploid potatoes through NAD metabolism, impacting multiple biological processes, including energy metabolism, hormone signaling, and macromolecular degradation. These findings not only advance our understanding of the molecular mechanisms underlying senescence but also provide a foundation for breeding potato varieties with tailored senescence timing to meet diverse agricultural and market needs.

## Figures and Tables

**Figure 1 ijms-26-04389-f001:**
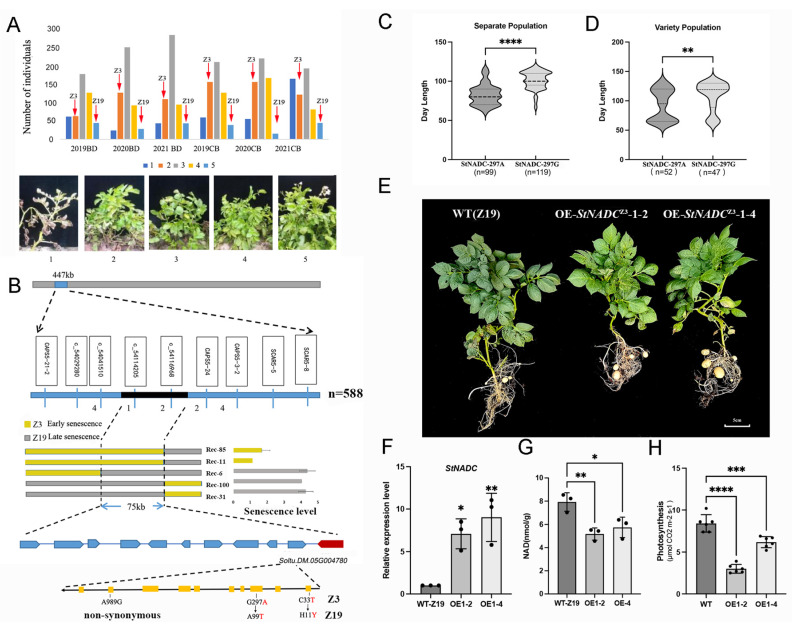
*StNADC^Z3^* allele promotes plant senescence in tetraploid potatoes. (**A**) Phenotypic identification of the plant senescence of the separate population. The plant senescence separate population of 588 progeny, early-senescence parent Z3, and late-senescence parent Z19 were evaluated in Baoding (BD) and ChaBei (CB) in 2019, 2020, and 2021, respectively. The plant senescence was assessed for each genotype from emergence to 50% senescence using the following scale: 5, dark-green plants; 4, light-green plants; 3, 25% of plant senescence; 2, more than 50% of plant senescence; 1, dead plant. (**B**) Fine mapping of the plant senescence locus on chromosome V narrows the locus between markers c5_4041510 and c5_4116968 to a 75 kb interval, a region that contains 11 predicted genes. The number below indicates the number of recombinants. The middle panel is the genotypes (**left**) and the senescence level of recombinant progeny (**right**). Bottom panel: the annotated gene structures of 11 candidate genes. The red nucleotide base or amino indicates the mutation in *Soltu.DM.05G004780* of Z3 compared to Z19. (**C**,**D**) The allele-specific KASP marker for *StNADC*-297A is closely related to senescence in the separate population (**C**) and variety population (**D**). (**E**) Plant growth of Z19 transformed with 35S::*StNADC*H-297A lines 1–2 and 1–4 after 45 days. The transgenic plants show earlier yellowing of leaves than the wild type. A minimum of six plants were considered for recording the observations. The experiments were repeated thrice with similar results. The scale bar was taken as 5 cm. (**F**) The relative expression levels of *StNADC* for the wild type Z19 and transformed lines. (**G**) NAD content per plant of wild type Z19 and *StNADC* transgenic plants, where the error bar indicates the standard deviation. (**H**) Photosynthetic rate of functional leaves measured with a CIRAS-3 instrument 45 days after planting (8:00 a.m. to 10:00 a.m.) in greenhouse conditions. **** *p* < 0.0001, *** *p* < 0.001, ** *p* < 0.01, * *p* < 0.05 for OE1–2 or OE1–4 compared with WT. The data were analyzed using one-way ANOVA.

**Figure 2 ijms-26-04389-f002:**
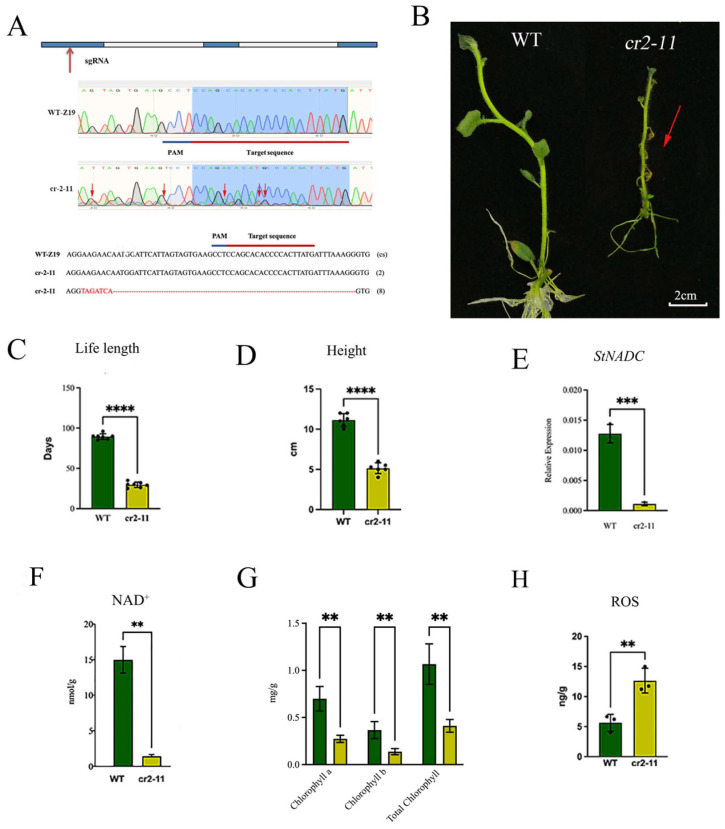
The *StNADC* gene plays a vital role in plant senescence and is indispensable for NAD synthesis. The green bar indicate the wilde type Z19, The yellow bar indicate *cr2-11*. (**A**) The gene structure of *StNADC* and the mutation patterns in T0 transgenic plants: sgRNA—single guide RNA; WT-Z19—wild type (tetraploid potato variety “Z19”); Cs—conserved sequence of *StNADC* gene in wild type. (2) and (8) represent the number of clones in mutant *cr2–11*. (**B**) The growth status of the wild type and *cr2–11* in MS medium at 30 days after propagation (DAP), The arrow indicates significant senescence of *cr2-11*. A minimum of 36 plants were considered for recording observations in each experiment. The experiments were repeated more than three times with similar results. The scale bar is 2 cm. (**C**,**D**) The life length (**C**) and height (**D**) of the wild type and *cr2–11*. (**E**) The relative expression levels of *StNADC* for the WT-Z19 and mutant *cr2–11*. The y-axis represents the *StNADC* expression level that is related to the internal control gene EIF-3e. The error bars represent the standard error of three biological replicates. (**F**) NAD content of wild-type and *cr2–11* mutant at 30 DAP. (**G**,**H**) Leaf contents of chlorophyll a, chlorophyll b, and total chlorophyll (**G**) and ROS (**H**) of wild type and *cr2–11* mutant at 30 DAP. The data of (**C**–**H**) indicate the mean values. Error bars refer to the standard deviation (SD) of three biological replications. **** *p* < 0.0001, *** *p* < 0.001, ** *p* < 0.01 or *cr2–11* compared with WT. The data were analyzed using one-way ANOVA.

**Figure 3 ijms-26-04389-f003:**
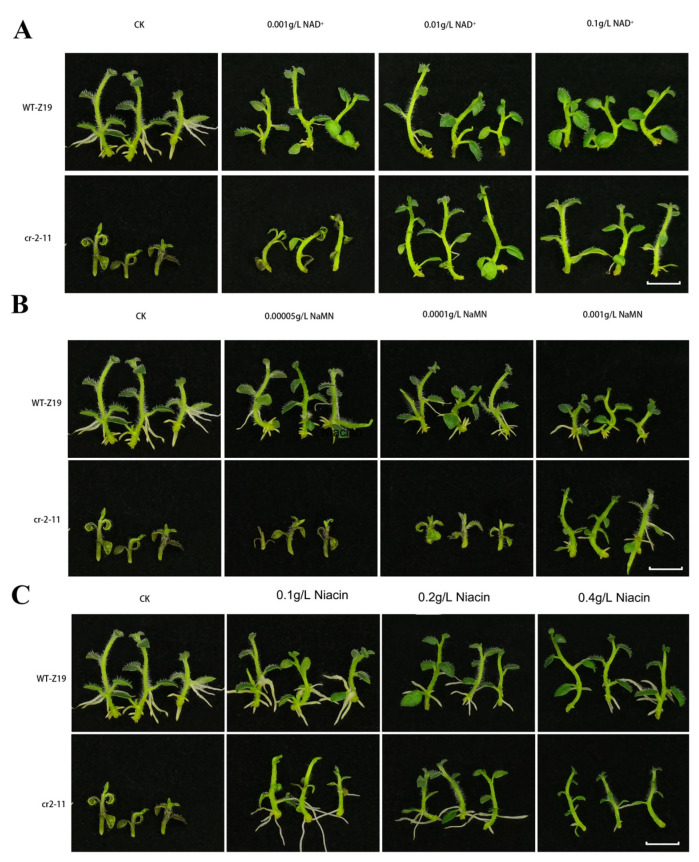
Appropriate concentrations of NAD+ and its intermediates in the NAD+ synthesis pathway can refresh *cr2–11* growth normally: (**A**) NAD, nicotinamide adenine dinucleotide. (**B**) NaMN, nicotinic acid mononucleotide. (**C**) Niacin, subtract of salvage pathway. CK, MS medium without niacin. The scale bar was taken as 1 cm.

**Figure 4 ijms-26-04389-f004:**
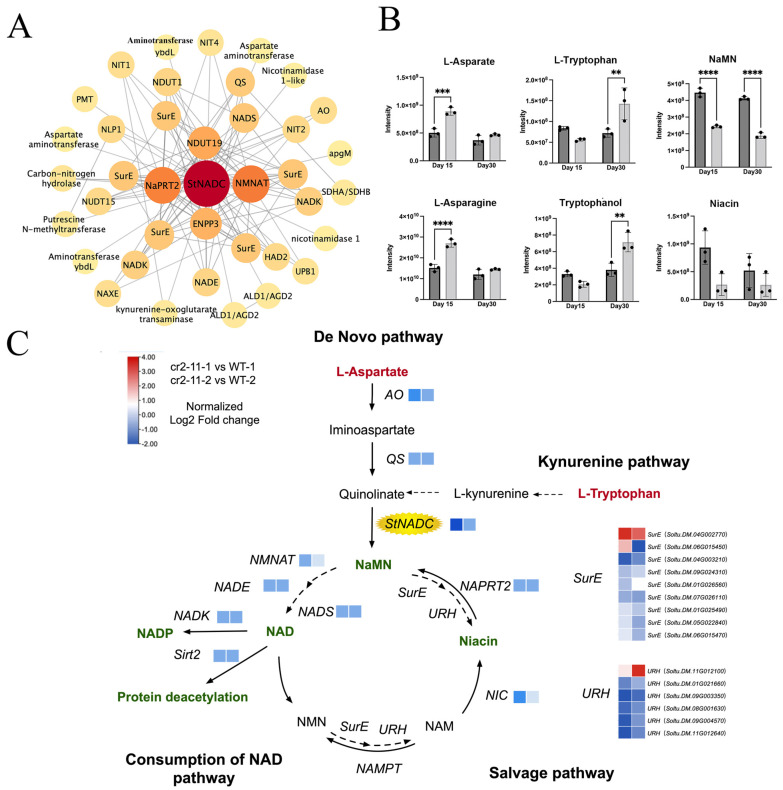
Knockout of *StNADC* has devastating effects on NAD synthesis and activates plant senescence and tuberization pathways. (**A**) Co-expression network of key genes in the NAD pathway. Pearson’s correlation coefficient values were calculated for each gene. The strength of the connection is represented by the size of the circle. (**B**) The metabolites related to *StNADC* in *cr2–11* and the wild type at Day 15 and Day 30 after propagation. **** *p* < 0.0001, *** *p* < 0.001, ** *p* < 0.01 (**C**) A simplified representation of the NAD biosynthetic pathway. The square heat map represents the transcriptional data related to NAD metabolism in transgenic (*cr2–11*) and WT plants. The normalized gene expression values [log2(FPKM+1)] were used to plot the heat maps.

**Figure 5 ijms-26-04389-f005:**
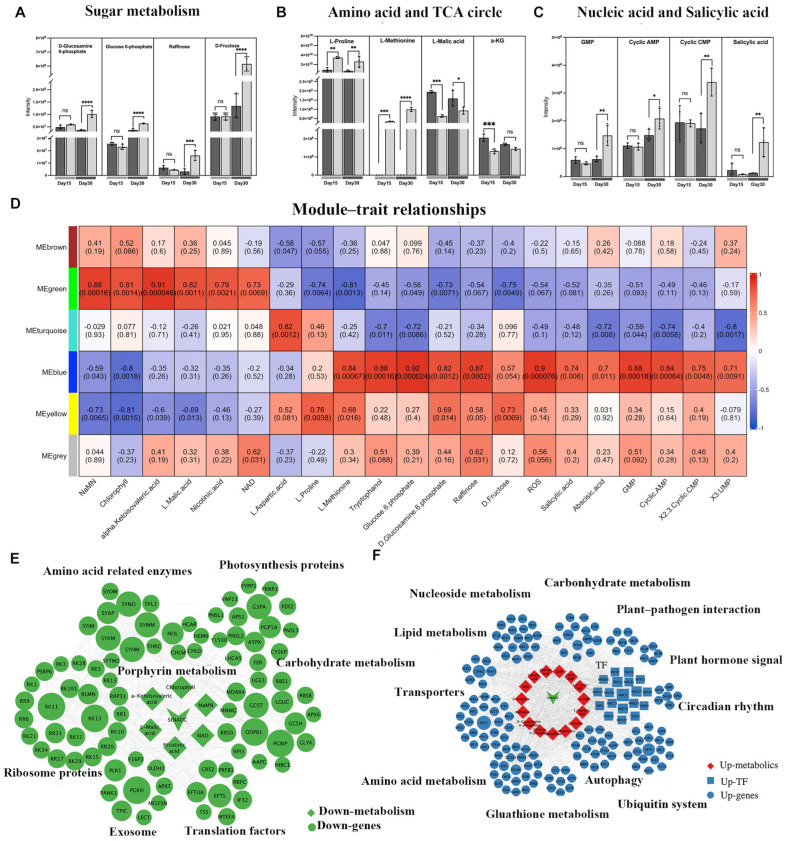
WGCNA reveals metabolic networks influenced by the *StNADC* gene. (**A**–**C**) Metabolic characteristic of *cr2–11* mutant. Relative metabolite contents of leaves at Days 15 and 30 are given together with change over time, **** *p* < 0.0001, *** *p* < 0.001, ** *p* < 0.01, * *p* < 0.05, (wild type: dark-gray bars; cr2–11: light-gray bars). Histograms show the relative amounts of soluble sugars (**A**), DNA damage intermediates and Salicylic acid (**B**), and amino acids and TCA circle intermediates (**C**). (**D**) Relationship between the modules and metabolites. (**E**) Co-expression network of downregulated genes module (green) with decline metabolites. (**F**) Co-expression network of upregulated genes module (blue) with elevated metabolites. Metabolites, structural genes, and TFs are marked in diamond, circle, and square shapes, respectively.

## Data Availability

The sequence of alleles of the *StNADC* gene is available in GenBank (accession number for *StNADC^Z3^*: OQ075559). RNA-seq data have been deposited in the NCBI with the accession numbers from SAMN32526031 to SAMN32526042. All other sequences from potatoes are available at GenBank or the Potato Genome Sequencing Consortium website (http://www.potatogenome.net). The Potato Genome Initiative locus numbers for the major genes are discussed in this article in Appendix A.

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
