# Peer review of "Natural Variation of StNADC Regulates Plant Senescence in Tetraploid Potatoes (Solanum tuberosum L.)"

_ijms, 2025, doi:10.3390/ijms26094389_

Round 1

Reviewer 1 Report

Comments and Suggestions for Authors

Abstract:

  • Gene names needs to be italicized
  • Rephrase: map based clone to map based cloning
  • No numbering in the manuscript

Introduction:

  • Which years’ fao data are authors using to report potato as third most important food crop after wheat and rice. How about maize?
  • What are critical enzymes authors are talking about?
  • Any reports of NAD+ role in senescence in plants? It will be relatable to see studies of NAD+ role in plants instead only in animals and plants
  • By which pathway NAD+ is synthesized in plants?
  • Mention the interval of the candidate region on chromosome 5
  • What generation is the progeny? F2?

Results:

2.1:

Why did authors screen F1 generation? In F1 generation, you can not look for segregation. F2 generation is the best generation where maximum segregation is achieved

Did authors screen all 588 plants over three years in both locations? Authors need to specify which generation of the population was tested where and in which year?

2.2:

Did authors identify the same QTL on chromosome 5 in this study as well? If yes, how did they identify that? What approach did authors use? Missing data/ information

2.3:

How did authors remove 10 other candidate genes in the 75Kb interval from the analysis. No information on that

Were the SNPs C33T and G297A deleterious?

2.4:

The greater transcriptional changes at Day 30 highlight the critical role of StNADC during senescence. How did authors reach that conclusion?

Why did authors randomly choose DEGs? Should it not have made more sense to choose genes involved in the NAD network?

Discussion:

3.2:

What does authors mean by El locus?

Author Response

Comments1: Gene names need to be italicized

Response1: We sincerely appreciate your careful reading. All gene names throughout the manuscript have now been italicized (line 15).

Comments2: Rephrase: map based clone to map based cloning

Response 2: We thank the reviewer for this observation. The text now uniformly uses 'map-based cloning'(line 16).

Comments3: No numbering in the manuscript

Response 3: We sincerely apologize for any inconvenience caused. Line numbers have now been added throughout the revised manuscript to facilitate review.

Introduction:

Comments 4: Which years’ fao data are authors using to report potato as third most important food crop after wheat and rice. How about maize?

Response 4: We appreciate the reviewer's insightful comment. Although maize (Zea mays) has a larger global cultivation area than potato (Solanum tuberosum), whereas over 60% of maize production is utilized for animal feed and industrial processing, so FAO classifies maize primarily as a grain/feed/industrial crop, whereas potato is explicitly categorized as the third most important food crop(especially as a staple in developing countries; see FAOSTAT 2023 report). 

Comments 5: What are critical enzymes authors are talking about?

Response 5: We sincerely apologize for any confusion caused. In response to reviewer’s comment, we have now included a detailed introduction of the key enzymes Quinolinate Phosphoribosyltransferase (QPT) ï¼ˆencode by StNADC ï¼‰(lines 75-79) to clarify their enzymatic functions.

Comments 6: Any reports of NAD+ role in senescence in plants? It will be relatable to see studies of NAD+ role in plants instead only in animals and plants

Response 6: We sincerely appreciate the reviewer's insightful comment on the role of NAD+ in plant senescence. As the reviewer rightly noted, we have expanded the evidence confirms NAD+'s crucial role in plant senescence in lines 79-84.

Comments 7: By which pathway NAD+ is synthesized in plants?

Response 7: We thank reviewer’s good question. In plants, NAD+ biosynthesis primarily occurs through the aspartate (de novo) pathway, with a secondary salvage pathway for recycling nicotinamide (NAM) (line75-77). 

Comments 8: Mention the interval of the candidate region on chromosome 5.  What generation is the progeny? F2?

Response 8: We sincerely apologize for any confusion regarding the genetic interpretation. Due to the heterozygous tetraploid nature of potato parental lines (2n=4x=48), the F1 progeny of potato crosses exhibit segregation patterns comparable to the F2 generation in inbred diploid crops (e.g., rice, maize). and we add this interpretation in materials and methods (line 390-392)

 Results: 

2.1:

Comments 9: Why did authors screen F1 generation? In F1 generation, you can not look for segregation. F2 generation is the best generation where maximum segregation is achieved.

Response 9: We appreciate the reviewer’s question. While unlike self-pollinated diploids (where F1 is uniform), potato F1 populations immediately segregate for traits, resembling an F2 population in diploids. For a simplex locus (Aaaa) in potato, F1 segregation approximates 1 (AAAA/AAAa):4 (AAaa):1 (Aaaa) — analogous to the 1:2:1 F2 ratio in diploids.
We have clarified this critical distinction in the revised manuscript (line405-408) .

Comments 10: Did authors screen all 588 plants over three years in both locations? Authors need to specify which generation of the population was tested where and in which year?

Response 10: We sincerely appreciate the reviewer's attention to data completeness. In our study, the F1 population comprised 588 progenies, with phenotypic data collected across multiple years and locations. As noted, some accessions had missing data points due to seed availability, germination failures. All missing data are explicitly documented in Supplementary Table S3: Summary of sample sizes per location/year.

 2.2:

Comments 11: Did authors identify the same QTL on chromosome 5 in this study as well? If yes, how did they identify that? What approach did authors use? Missing data/ information

Response 11: We appreciate the reviewer's insightful comments regarding our mapping approach. To refine the candidate region on chromosome 5, we expanded the population to 588 F1 progenies, enhancing recombination resolution, and developed novel markers (KASP/SNPs) across the target interval and identified 5 critical recombinants reduced the interval to 75kb region containing 11 candidate genes.

All data, including missing phenotypic records annotated in Supplementary table S1.

Comments 12: How did authors remove 10 other candidate genes in the 75Kb interval from the analysis. No information on that.

Response 12: We acknowledge reviewer’s insightful question. Senescence is a polygenic trait controlled by multiple pathways.  The prioritized gene StNADC were selected based on NAD pathway function in senescence in both animal and plants. And functional studies of few other candidates in Supplementary Table 3 are ongoing, and their roles will be reported in future work. We have now clarified this limitation in the Discussion (line 382-386).

Comments 13: Were the SNPs C33T and G297A deleterious?

Response 13: We thank the reviewer for highlighting this important point. Based on functional evidence, overexpression of the mutant alleles in potato accelerated senescence (Fig. 1B), while overexpression of the wild-type alleles showed no observable effect. We confirm that both C33T (H11Y) and G297A (A99T) are deleterious mutations

2.4: 

Comments 14: The greater transcriptional changes at Day 30 highlight the critical role of StNADC during senescence. How did authors reach that conclusion?

Response 14: We sincerely appreciate the reviewer's valuable feedback regarding the interpretation of StNADC's role in senescence. In response to this comment, we have modified the conclusion about StNADC's function by removing the phrase "highlight the critical role" from the manuscript(line 245).

Comments 15: Why did authors randomly choose DEGs? Should it not have made more sense to choose genes involved in the NAD network?

Response 15: We appreciate the reviewer’s insightful question regarding gene selection criteria. Our choice of DEGs randomly for qRT-PCR validation of the reliability of the RNA-seq data.

 Discussion:

3.2: 

Comments 16: What does authors mean by EI locus?

Response 16: We appreciate the reviewer’s careful reading,The EI locus (Early Inactivation locus) is one of the earliest genetically defined senescence-related loci identified in potatoes (Solanum tuberosum),with premature leaf senescence (early yellowing and reduced photosynthetic activity before natural maturity,and we have add its meaning in line 329.

Reviewer 2 Report

Comments and Suggestions for Authors

Please find the attachment with detailed suggestions for correcting the manuscript.
The manuscript "Natural variation of StNADC regulates plant senescence in Tetraploid Potatoes (Solanum tuberosum L.) " is interesting and based on well-designed research. However, a few issues need to be addressed to improve it. 
The introduction is quite short; there is a lack of aging context in plants and a lack of information about the mechanisms of plant aging related to redox molecules. The Result section is somewhat confusing. The Authors should explain what they mean by writing about senescence and aging. Are those terms equal in the manuscript? 
The Materials and Methods and the Results sections are well-described. Some information from the Discussion section should be moved to the introduction. Besides these suggestions, the manuscript will be an interesting research for Plants readers.

Author Response

Comments 1: Please find the attachment with detailed suggestions for correcting the manuscript.

Response 1: We are grateful to the reviewers for their constructive feedback. Each formatting concern highlighted in the supplementary materials has been corrected, with track changes .

Comments 2: The manuscript "Natural variation of StNADC regulates plant senescence in Tetraploid Potatoes (Solanum tuberosum L.) " is interesting and based on well-designed research. However, a few issues need to be addressed to improve it. The introduction is quite short; there is a lack of aging context in plants and a lack of information about the mechanisms of plant aging related to redox molecules.

Response 2: We sincerely appreciate your valuable suggestion. In response, we have expanded the introduction to include: (1) the context of senescence in plants(line 35-44), and (2) mechanisms of plant aging related to redox molecules (line 56-60). These additions provide important conceptual grounding for our study.

Comments 3: The Result section is somewhat confusing. The Authors should explain what they mean by writing about senescence and aging. Are those terms equal in the manuscript? 

Response 3:We sincerely apologize for any confusion caused by the inconsistent terminology. In this manuscript, 'aging' and 'senescence' carry the same meaning. The terminology has been consistently updated to 'senescence' throughout the manuscript to maintain technical accuracy.
Comments 4: The Materials and Methods and the Results sections are well-described. Some information from the Discussion section should be moved to the introduction. Besides these suggestions, the manuscript will be an interesting research for Plants readers.

Response 4: We thank the reviewer for this helpful observation. Relocating portions of the discussion to the introduction(line 35-44) has significantly improved the paper's logical structure.

Reviewer 3 Report

Comments and Suggestions for Authors

Authors identified Nicotinate-nucleotide pyrophosphorylase QPT/StNADC as a novel central senescence regulator. Overexpression of the StNADC decreased the NAD content and accelerated senescence in late senescence variety. Similar results are obtained in plants containing mutation of StNADC,induced by  CRISPR/Cas9. Supplementaion of NAMN, NAD or niacin can rescue the plant mutated phenotype. Results obtained by Authors show the pivotal role of StNADC in NAD synthesis, which could be applied to regulate senescence in potato through targeting the StNADC-mediated NAD synthesis pathway. Study is correctly planned and performed, obtained results support conclusions. Study could be interesting to researchers in the field. Following improvements should be included into the manuscript:

  1. Fig 1 B,E, Fig 4 and 5. There are problems with reading words written in a small font, try to increase the font size (or bolding), alternatively make separate Figures to increase font and picture size.
  2. Section 2.1.3 and Fig 1 S3.

In the sentence: Real-time PCR analysis revealed that StNADC is expressed in roots, stems, leaves, and flowers (Figure S1.A )….. , write the StNADC in italics as a gene name.

If possibile, based on available references, add information related to molecular changes in active site or other parts of enzyme as a result of polymorphisms and amino acid residua substitution found by Authors. If these informations could be found in references, discuss them in the discussion section.

  1. Section 2.1.4

Italicize Nicotiana benthamiana as a plant Latin name.

  1. Section 3.3

Italicize Escherichia coli

  1. Section 4.5

Italicize Agrobacterium tumefaciens

  1. Section 4.11

Italicize S. tuberosum

  1. Section 4.11

Add volume and concentration of libraries.

Add name of software used to align obtained sequences to reference genome.

Provide units (FPKM) to Express the RNA frequency.

  1. tuberosum write in italics
  2. Correct typographical errors as joined words in the Supplementary information Word file.
  3. Try to discuss based on own and available reference why the both: overexpression and misfunctional mutation of StNADC leads to the decreased NAD concentration.
  4. Results of RNAseq experiments (for example raw data) should be submitted to batabases as those at NCBI, to obtain the accession number. This number should be presented in the article.

Comments on the Quality of English Language

Correct the small typographical errors mentioned in the review.

Author Response

Comments 1: Fig 1 B, E, Fig 4 and 5. There are problems with reading words written in a small font, try to increase the font size (or bolding), alternatively make separate Figures to increase font and picture size.

Response 1: We sincerely appreciate this constructive suggestion. As recommended, we have now enlarged and bolded the fonts in the following figures: Figure 1B, 1E,Figure 4 and Figure 5. The modified figures are included in the revised manuscript . We believe these changes significantly enhance the clarity of the data presentation.

Comments 2: Section 2.1.3 and Fig 1 S3.

In the sentence: Real-time PCR analysis revealed that StNADC is expressed in roots, stems, leaves, and flowers (Figure S1.A )….. , write the StNADC in italics as a gene name.

Response 2: We sincerely appreciate the reviewer’s careful reading and valuable suggestions. As recommended, we have corrected all gene symbols to italic format throughout the manuscript.

Comments 3: If possible, based on available references, add information related to molecular changes in active site or other parts of enzyme as a result of polymorphisms and amino acid residua substitution found by Authors. If these information could be found in references, discuss them in the discussion section.

Response 3: We sincerely appreciate the reviewer's valuable suggestion. Our study identified three key SNPs (C33T, G297A, and A938G) in StNADC between Z3 and Z19, resulting in two amino acid substitutions (H11Y and A99T) and a non-synonymous mutation (A938G). We have expanded the Discussion as follows:

The H11Y substitution (C33T) replaces a positively charged histidine with polar tyrosine in the N-terminal domain, likely disrupting α-helix hydrogen-bond networks critical for structural stability and potentially altering sub-strate/cofactor electrostatic interactions, as observed in Bacillus subtilis NADH dehydro-genase variants [35]. And the A99T substitution (G297A) introduces a hydroxyl group near the active site, potentially rigidifying the active-site loop through new hydrogen bonds while possibly causing steric hindrance to substrate binding, similar to Pseudo-monas NAD kinase mutations [36].(line346-356)

Comments 4: Section 2.1.4 Italicize Nicotiana benthamiana as a plant Latin name.

Response 4: We sincerely appreciate the reviewer’s careful reading, as recommended, we have italicized Nicotiana benthamiana throughout the manuscript (including Line 177 and all other instances).

Comments 5: Section 3.3 Italicize Escherichia coli.

Response 5: We sincerely appreciate the reviewer’s careful reading, as recommended, we have italicized Escherichia coli in line 360.

Comments 6: Section 4.5 Italicize Agrobacterium tumefaciens

Response 6: We sincerely appreciate the reviewer’s careful reading, as recommended, we have italicized Agrobacterium tumefaciens in line 454.

Comments 7: Section 4.11 Italicize S. tuberosum

Response 7: We sincerely appreciate the reviewer’s careful reading, as recommended, we have italicized S. tuberosum in line 529.

Comments 8: Section 4.11 Add volume and concentration of libraries.

Response 8: We sincerely appreciate this valuable suggestion. In the revised manuscript, we have added explicit experimental details in the Methods section (Section 4.11, line 522-524 ). Final libraries were prepared in a volume of 20 μL and diluted to a standardized concentration of 2 nM using EB buffer (10 mM Tris-HCl, pH 8.0) for sequencing.

Comments 9: Add name of software used to align obtained sequences to reference genome.

Response 9: We sincerely appreciate the reviewer's insightful suggestion. We use the blast tool of potato reference genome website (https://spuddb.uga.edu) to align obtained sequences with default parameters.

Comments 10: Provide units (FPKM) to Express the RNA frequency.

Response 10: We sincerely appreciate the reviewer’s careful reading and constructive feedback. The processed RNA-seq data, including FPKM values, are provided in he NCBI GEO/SRA submission (Accession: SAMN32526031 to SAMN32526042) in Data Availability Statement section (line 599).

Comments 11: tuberosum write in italics

Response 11: We sincerely appreciate the reviewer’s careful reading, as recommended, we have italicized tuberosum throughout the manuscript.

Comments 12: Correct typographical errors as joined words in the Supplementary information Word file.

Response 12: We sincerely appreciate the reviewer's careful reading and valuable feedback. We have systematically reviewed the entire Supplementary information Word file to identify all typographical errors and correct them in revised file. These corrections have significantly improved the clarity and professionalism of our Supplementary Information. We appreciate the opportunity to enhance our manuscript's quality.

Comments 13: Try to discuss based on own and available reference why the both: overexpression and misfunctional mutation of StNADC leads to the decreased NAD concentration.

Response 13: We sincerely appreciate the reviewer's insightful suggestion. As noted, Overexpression of the StNADCZ3 (a hypomorphic allele with partial loss-of-function), and CRISPR/Cas9-mediated knockout of StNADC both resulted in reduced NAD+levels. This aligns with our hypothesis that the StNADCZ3 mutation (H11Y/A99T) likely disrupts enzyme stability, while Complete knockout abolishes NAD+ salvage synthesis. We have added this discussion in lines 346-356.

Comments 14: Results of RNAseq experiments (for example raw data) should be submitted to databases as those at NCBI, to obtain the accession number. This number should be presented in the article.

Response 14: We sincerely appreciate the reviewer's suggestion regarding data availability. We have officially deposited all RNA-seq data in NCBI's Gene Expression Omnibus (GEO) under accession number: SAMN32526031 to SAMN32526042. Updated the manuscript (Lines 598-599) to explicitly state: "RNA-seq data have been deposited in the NCBI with the accession number SAMN32526031 to SAMN32526042. "

Round 2

Reviewer 3 Report

Comments and Suggestions for Authors

Authors corrected the manuscript according to suggestions, I have no other comments.